# Collaboration of Drone and Internet of Public Safety Things in Smart Cities: An Overview of QoS and Network Performance Optimization

**Saeed H. Alsamhi [1,2,*], Ou Ma [3], M. Samar Ansari [4]** and **Sachin Kumar Gupta [5]**

1   School of Aerospace Engineering, Tsinghua University, Beijing 100084, China
2   Department of Electonics Engineering, IBB University, Ibb, Yemen
3   College of Engineering and Applied Science, University of Cincinnati, Cincinnati, OH 45220, USA; ou.ma@uc.edu
4   Software Research Institute, Athlone Institute of Technology, Athlone, Ireland; mdsamar@gmail.com
5   Department of ECE, SMVDU, Katra, (Jammu & Kashmir), India; sachin.rs.eee@itbhu.ac.in
*   Correspondence: salsamhi@tsinghua.edu.cn; Tel.: +86-178-0123-9122

**Abstract:** This paper studies the network performance of collaboration between the Internet of public safety things (IoPST) and drones. Drones play a vital role in delivering timely and essential wireless communication services for the recovery of services right after a disaster by increasing surge capacity for the purposes of public safety, exploring areas that are difficult to reach, and providing an area of rapid coverage and connectivity. To provide such critical facilities in the case of disasters and for the purposes of public safety, collaboration between drones and IoPST is able to support public safety requirements such as real-time analytics, real-time monitoring, and enhanced decision-making to help smart cities meet their public safety requirements. Therefore, the deployment of drone-based wireless communication can save people and ecosystems by helping public safety organizations face threats and manage crises in an efficient manner. The contribution of this work lies in improving the level of public safety in smart cities through collaborating between smart wearable devices and drone technology. Thus, the collaboration between drones and IoPST devices establishes a public safety network that shows satisfying results in terms of enhancing efficiency and information accuracy.

**Keywords:** IoT; public safety network; drone; emergency communication; IoPST; SAR; smart cities

## 1. Introduction

Recently, drones have become attractive for research and a variety of applications, because of their flexibility and the possibility of use in a wide range of applications, such as the security, control, monitoring, and exploration of terrestrial areas otherwise difficult to reach quickly. Furthermore, this is transformative technology, enhancing how first responders can reach and carry out rescue missions at the sites of natural disasters. In addition, it can provide support for delivering medical supplies, as well as for emergency management cases such as forest firefighting, critical infrastructure protection and inspection, coastal monitoring, and police augmentation, in addition to helping smart cities meet their public safety requirements. The relationship between public safety and the Internet of Things (IoT) was discussed in [1], while a taxonomy of the IoT-based smart city was provided in [2]. Applying IoT technologies to smart cities could lead to changes and improvements in the economy, safety, management of public utilization, and transportation in smart cities.

The most attractive application of drones is the collection of data from IoT using wearable devices in smart cities or at events [3,4]. A drone can communicate with heterogeneous devices on the ground that are unable to communicate over large areas and can send information to the intended destination

in order to make a final decision and take action in a timely fashion. The development of such communication technologies will lead to changes in how we communicate with the vast number of things connected to the Internet. However, the IoT refers to the connections among the things themselves. The concept of IoT refers to things that are not intelligent, and does not include artificial intelligence (AI). Furthermore, the Internet of robotic things (IoRT) is an intelligent concept that gives the associated things the capacity for negotiation, reasoning, decision, and delegation [5]. In contrast, the Internet of public safety things (IoPST) refers to wearable things for first responders that are used for public safety and are connected via the Internet, such as cameras, sensors, and drones. IoPST covers different types of public service, such as emergency response, disaster relief, traffic control, crime control, investigating accidents, smart event management, etc., as shown in Figure 1.

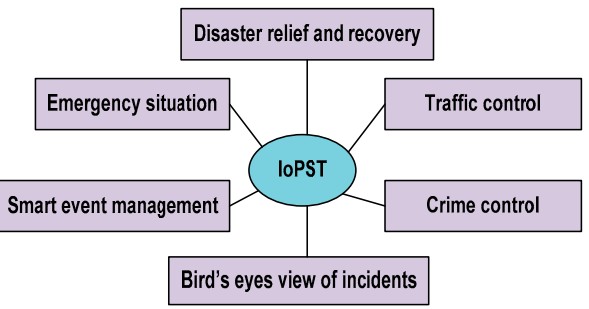

**Figure 1.** An illustration of the IoPST concept.

Wearable IoPST devices connect with each other in smart cities for the purposes of public safety, and allow first responders to share information with command posts and relevant parties. These connected devices can help relief teams and police to identify when or where disasters or crime have happened, as shown in Figure 2. On the basis of this, the police can successfully take decisions in order to track down perpetrators.

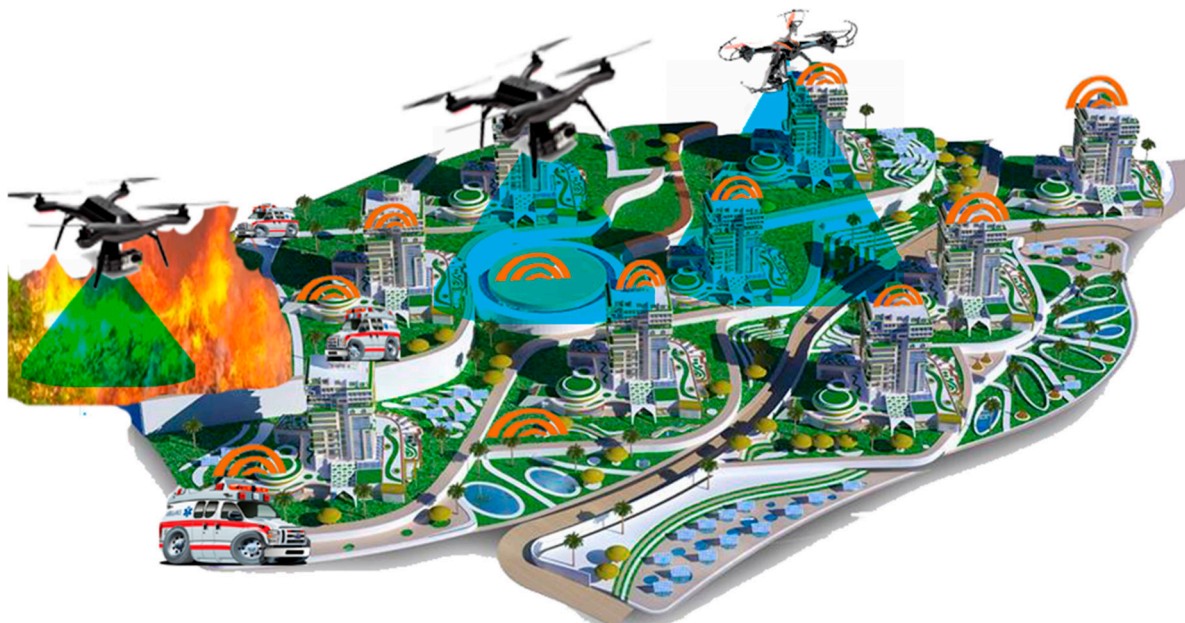

**Figure 2.** An illustration of devices connecting to a drone for the public safety network.

However, due to the limited transmission power of wearable devices, they are unable to send data over long distances. Drone technology, however, is able to deliver services over a large geographical zone in real time. The use of drone technology represents the cheapest and most efficient way to

monitor a criminal on the run, find a missing person, survey a disaster scene, etc., especially in time-critical situations. Therefore, the best features of the integration of drones and IoPST are those that are supposed to achieve and provide maximum benefits, such as their reprogrammability, good sensing capability, ability to interconnect and identify things, ubiquity, communication capability, etc.

This article mainly focuses on the network performance of drones and IoPST. Furthermore, we introduce the novel concept of collaboration between drones and IoPST in smart cities, in order to enhance the network performance and maintain the QoS. Furthermore, we discuss drones in public safety applications. With respect to public safety, we discuss the capability of drones and IoPST to collect data in real time, track crimes, and guide police in finding crime locations, as well as guiding Search and Rescue (SAR) teams effectively and efficiently, leading to improved orientation [6]. In this case, the collaboration between drone technology and IoPST represents a key technology for determining the location of lost people in SAR events in smart cities. Additionally, metrics for the evaluation of the performance of drones and IoPST are considered, including path loss probabilities, elevation angle, delay, and throughput. Our results show that the collaboration of drones and the IoPST can significantly enhance the performance of emergency communication services by maintaining high QoS, helping SAR teams to perform their duties efficiently, and reducing economic losses.

## 1.1. Related Work

Due to the occurrences of natural disasters, crime, and terrorist attacks, emergency services and healthcare providers, in particular, must pay close attention to safety [7]. For this purpose, the wireless communication linkage plays a vital role in assessing a damaged area, collecting data on supplies, helping police to find the locations of incidents in smart cities, coordinating rescue team and relief team activities, saving people's lives, and accounting for missing people. To establish an efficient and effective wireless communication network for delivering data in a disaster area spectrum width is highly recommended [8,9]. However, terrestrial wireless communication technologies could be missing, unavailable through congestion, or damaged [10]. The critical factors for emergency communication network solutions and disaster recovery are rapid deployment, immediate availability, and reliability.

Therefore, space technologies represent the best solution for disaster recovery, public safety, SAR, and emergency services [11]. Space technologies are used for collecting information needed to protect human and reduce economic losses. In the aftermath of a disaster, a satellite is a reliable communication solution [12], but the weakness is a time delay and launching cost. Therefore, using an aerial platform can be a better solution, because it has the merits of both space and terrestrial wireless communication systems. Mohorcic et al. [13] argued the use of aerial platforms for disaster and emergency situations and also showed the significance of rescue teams during the disaster. The ability of an aerial platform to deliver communication services such as E911 to facilitate SAR operations is discussed in [14]. The advantages of aerial platforms are the capability and stable coverage area, survivability [15], mitigated interference that occurs in the wireless communication [16], and ability to manage traffic [17]. They offer a valuable alternative to support emergency communications after a disaster [18]. Disaster prediction coverage for mitigation of disaster impact a from low-altitude platform (LAP) is discussed in [19].

Drones belong to the LAP family [20,21], and are considered to be space robots. The significant advantages of drones are deployment cost, line of sight (LoS), low propagation delay, rapid deployment, fixed station, and use in disasters. The propagation models, mobility, and positioning of drones for a communication network are discussed in [22]. Drones are attractive for emergency communication because of the possibility of rapid deployment and users operating them from their existing mobile handsets in disaster zones. Therefore, drones represent the best solution for disaster recovery and emergency services because they can be used to support relief and rescue teams in performing their tasks efficiently. Drones play a vital role in connected devices in smart cities. Hence, the authors of [23] reviewed the various aspect of drones related to privacy, cyber security, and public safety in smart cities.

IoT devices, robots, and humans can communicate as a cooperative. In particular, when flying drones are used, they can support the connectivity of existing terrestrial wireless networks such as cellular and broadband networks. Compared to terrestrial base stations, the advantage of using drones as flying base stations is their ability to provide on-the-fly communications and to establish a LoS communication links to ground users. However, the coverage area for the deployment of drone base station was considered for minimizing energy cost, improving the coverage radius and optimal altitude of drones [24]. Indeed, another important application of drones is in IoT scenarios [25,26], because devices often have small transmission power and may not be able to communicate over a long range. Sharma et al. [27] summarized the various cooperation approaches for the formation of drones. Also, Sharma et al. [28] developed a scenario for traffic management and cooperation of drones and nodes on the ground to provide continuous data transfers and network stabilization by using adhoc technology. The target applications and technological implications of IoT-aided robotics were discussed [29]. Furthermore, Dutta et al. [30] addressed the network security enhancement of IoT-aided robotics in a complex environment. In addition to the work of the authors in [30], the authors of [31,32] reviewed the convergence in terms of network protocols, architectures, and embedded software for IoT robotics for smart cities. The interaction of robotic and IoT devices was investigated [33,34]. Therefore, AI, robots, and IoT will provide the next generation of IoT applications [35]. Moreover, renewable energy harvesting was discussed for the energy Internet of drone communication and networks [36]. Therefore, drone technology and information and communication technologies pay a vital role in smart cities' reduction of resource consumption and costs. Greening information and communication technologies enabled the green IoT [37] by reducing energy consumption, pollution, and hazardous emissions in smart cities. Most of the challenges regarding energy efficiency, interference, and communication networks are discussed in depth [38–42], along with intelligent techniques for processing data [43,44].

Furthermore, the authors of [45] explored the potential usefulness of the IoT to enhance public safety and discussed the challenges and opportunities of using the IoT to support public safety networks and SAR. The hybrid network architecture for public safety broadband communication was discussed in [46], in which the stationary base station supports heavy traffic after an incident. However, LTE network architecture towards 5G in order to support emerging public safety was discussed in [47]. The authors of [48] analyzed how the deployment of the drone as a base station could deliver communication services to a particular area. Furthermore, the coexistence between heterogonous devices and drones was discussed. The fundamental operation and techniques were introduced for enhancing the efficiency and accuracy of public safety network using IoT technologies. Moreover, the authors of [49] reviewed the techniques and the availability of IoT for disaster management. Reina et al. [50] outlined the disaster management and the importance of using IoT and big data. The study focused on the use of drones as $0^{th}$ responders to deliver communication service to victims in a disaster. The main idea of the $0^{th}$ responder was to arrive at the disaster area before the first responders. The work was divided into two parts, the collection of data and the use of local searching to find the optimal position in which the drone could deliver communication services to the victims. Furthermore, the work in [50] was supported by [51], in which the position of the $0^{th}$ responder was considered for delivering communication services to victims in disaster areas. However, keeping a connectivity link between the $0^{th}$ responder and other responders was not discussed. Therefore, Alsamhi et al. [52] developed an artificial neural network to predict signal strength between the drone and wearable IoT devices or other responders on the ground to keep up connectivity for delivering services. The collaboration of drones and IoT plays a significant role in public safety. The proposed network architecture, that is, the integration of drones, IoT and smart wearable devices, offers numerous services like supporting disaster relief team to save human lives, long-distance communication, greening communication, etc. To the best of our knowledge, no study has been done to evaluate the collaboration of drone and IoPST performance network. The summary of related work is shown in Table 1.

**Table 1.** Summary of the related work.

| Ref. | Highlighted | SAR | IoT | Drone | Smart Cities | Public Safety | Focus |
|---|---|:---:|:---:|:---:|:---:|:---:|---|
| Reina et al. [50] (2018) | Importance of using IoT and big data for disaster management and public safety | √ | √ | √ | | √ | ○ Use of drones as 0th responders to deliver communication service to victims<br>○ Collection of data and the use of local searching to find the optimal position |
| Mozaffari et al. [48] (2016) | Drone deployment to provide wireless communications services to a given area | √ | √ | √ | √ | | ○ Co-existence between drone and D2D communication network<br>○ The tradeoff between coverage and delay |
| Zanella et al. [32] (2014) | IoT for smart cities | | √ | | √ | | ○ Reviewed the enabled protocols, technologies, and architecture of IoT for smart cities. |
| Alsamhi et al. [37] (2018) | Green IoT for healthy environment and in smart cities | √ | √ | √ | √ | | ○ Reviewed the enabling technologies to reduce pollution, energy consumption, and resource usage. |
| Msadaa et al. [45] (2016) | IoT in Support of Public Safety Networks | √ | √ | - | - | √ | ○ Increased global connectedness to be offered by IoT<br>○ Save lives and the environment by helping PS organizations face threats and manage crises efficiently |
| Mehmood et al. [2] (2017) | Devise a taxonomy of the IoT-based smart city | ○ √ | ○ | | ○ √ | ○ | ○ Source IoT platforms for realizing smart city applications |
| Chen et al. [46] (2016) | Average traffic of public safety networks architecture | ○ √ | ○ | ○ | ○ | ○ √ | ○ Reduction of the required number of base stations |
| Favraud et al. [47] (2016) | LTE network architecture towards 5G in order to support emerging PS | ○ | ○ | ○ | ○ √ | ○ √ | ○ Supporting network mobility<br>○ Less expensive |
| Naqviet al. [53] (2018) | Importance of incorporating the drones in the heterogeneous network for capacity and coverage extension | √ | | √ | | √ | ○ Enable high data rate<br>○ Improving the 5G energy efficiency |
| Kolios et al. [6] (2016) | The communication technologies that enable data dissemination in public safety networks | √ | | | | √ | ○ Effective and efficient data dissemination represents the key technology of processing public safety.<br>○ Collected data help to build a better picture of the situation<br>○ Data distributed to individuals can reassure them of the measures taken to alleviate the risks. |
| Mohamed et al. [4] (2018) | Integrating drones in smart cities | | √ | √ | √ | √ | ○ cost-effective services<br>○ reduce resource consumption |

**Table 1.** *Cont.*

| Ref. | Highlighted | SAR | IoT | Drone | Smart Cities | Public Safety | Focus |
|---|---|---|---|---|---|---|---|
| Reina et al. [51] (2018) | Drone as 0th responders and its best position | √ | √ | √ | | √ | ○ 0th responders are to arrive at the disaster scenario before the 1st responder<br>○ Finding the best position of 0th responders |
| Alsamhi et al. [52] (2018) | Signal strength between drone and IoT device in smart cities | | √ | √ | √ | | ○ Predict the accurate signal strength |
| Alsamhi et al. [The proposed work] | Collaboration of drone and Internet of public safety things | √ | √ | √ | √ | √ | ○ The collaboration of advanced technologies for public safety<br>○ Efficient connectivity services<br>○ Enhancing QoS |

## 2. Public Safety Network

A public safety network between victims and first responders is vital for search and rescue operations. Capable and fast emergency communication is needed for an effective public safety communication network. During public safety communication, efficient communication technology will save people's lives and improve connectivity among the SAR teams. WiMAX, WiFi, ad hoc, and long-term evolution (LTE) are the potential broadband communication types dedicated for public safety networks. However, these technologies are useless if a disaster occurs. Therefore, space technology is the best solution for public safety and disaster recovery when terrestrial communication systems are destroyed [54]. In this regard, drones play a vital role in public safety and disaster recovery/emergency services because they are able to reach areas human rescue teams cannot reach immediately and provide a critical first response.

### 2.1. Drones for Public Safety

Drones can fly autonomously in the sky and are associated with different applications in civilian tasks such as transportation, communication, agriculture, disaster mitigation, emergency response, smart things, and environmental preservation. They are a promising technology because of their rapid and easy deployment, ability to dynamically change location in an emergency situation, quick reconfiguration, and flexible technology. Furthermore, they can provide effective communication for a public safety network, as shown in Figure 2. Moreover, drones can move around to provide large disaster coverage area faster, and achieve ubiquitous connectivity within a minimum time in a public safety network.

Emergency communications are the most important and specialized field for giving high value to people's lives. During an earthquake or tsunami, there is no ground transportation, so drones can provide and maintain wireless emergency communication services during and after the disaster. Therefore, the SAR team can easily perform their tasks in a sequential manner. When a drone reaches the desired altitude, the affected area will be surveyed via a digital video camera. Then, collected images will be sent to a ground station for monitoring the area of operation and coordinating SAR teams with first responders' arrival. The authors of [40] introduced a multi-drone network architecture and demonstrated the network benefits in SAR and disaster assistance.

Drones are all about the payload, so the type of event determines what equipment should be in the payload. Therefore, the most common drones used for public safety are the Dà-Jiāng Innovations (DJI) Inspire and DJI Phantom, as shown in Figure 3; both are from the DJI family. The DJI Phantom is easy to acquire, land and take off, flexible in terms of the camera tilting, and able to fly around objects, but even experienced pilots can get into tough spots. The features and advantages of the DJI drone are explained in [23]. Furthermore, DJI drones provide better SAR [55], so they can be used

for all sorts of activities such as environmental monitoring, disaster management, mapping, and 3D modeling. During the launch of a DJI drone, it starts looking for interference, and when found it will try to localize the interferer. It will also change the flight plan to get the next meaningful measurement point, reducing the flight time. The DJI drones are efficient at capturing imagery of sufficient quality for the 3D mapping required for accident reconstruction and crime scene visualization in smart cities.

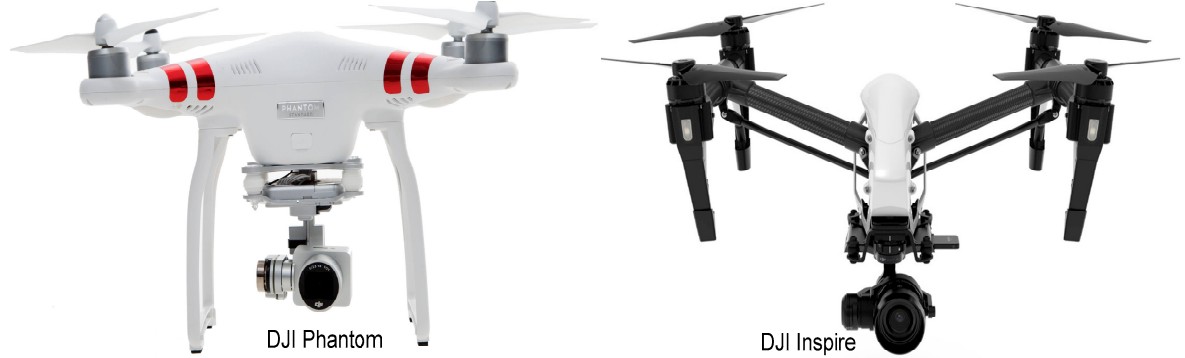

**Figure 3.** Public safety DJI drone [23].

Recently, a lot of studies have investigated various design challenges such as 3D deployment, energy efficiency, and time flight constraint. The optimal 3D deployment of drones for extending the coverage area and enhancing the QoS is discussed in [56]. The idea is supported by a study [57] that discusses the 3D deployment of a multiple drone base station to maximize the coverage area and maintain the link quality between drones and the ground station by using the practical swarm optimization technique. Furthermore, drones' 3D deployment represents a key technology that can assess drones to deliver network services for public safety and disaster management. Therefore, the authors of [58] proposed a practical swarm optimization technique to enable the drones' 3D deployment to provide a large coverage area, maintain connectivity, and satisfy users' QoS requirements and drone capacity. Also, 3D deployment for swarm drones was introduced for maximizing the available lifetime of drones and the total throughput of all users [59].

2.1.1. Coverage Area

The optimal placement of drones for public safety communications to enhance the coverage performance is discussed in [60]. The coverage can be extended with a number of drones, as shown in Figure 4. These drones can connect with each other and with first responders over an emergency or disaster area for delivering services accordingly. A drone lifts a flying platform by incorporating communication technologies such as ad hoc, WiFi, WiMAX, or LTE equipment to provide efficient communication services. A disaster generally destroys all of the communication and electric supply infrastructure. For some people, the availability of electricity means the difference between life and death. Electric power is required for lighting medical aid stations, ventilation, and delivering water to people in shelters, as shown in Figure 5. The figure illustrates the architecture power beaming transport from a drone. Therefore, power is beamed from a drone to the rescue and relief team receiver devices over the disaster area.

During an earthquake, power may go out immediately. During a tsunami, the water level remains high for several hours, and many people are swept away several kilometers out to sea, creating an urgent need for rescue and relief over the sea. Therefore, the emergency communication system must take into consideration that no support can come via ground transportation. Hence, power transmission technologies must have a time-varying electromagnetic field [61]. Wireless transmission power is electric power that transmits from the power source to destination machines without discrete human-made conductors [62]. Wireless power transmission is useful in case of hazards. A transmitter machine connected to the source of power, then using an electromagnetic field for power transmission

across an intervening space to one or more receivers, will finally be converted back to electric power for utilization [61].

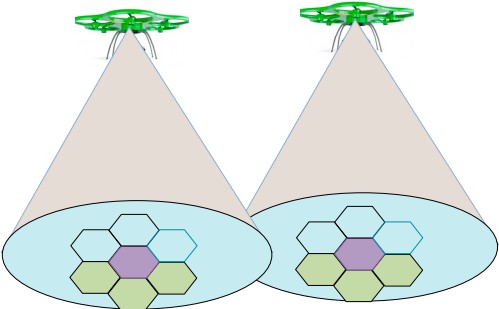

**Figure 4.** An extension of the public safety network by using a team of drones.

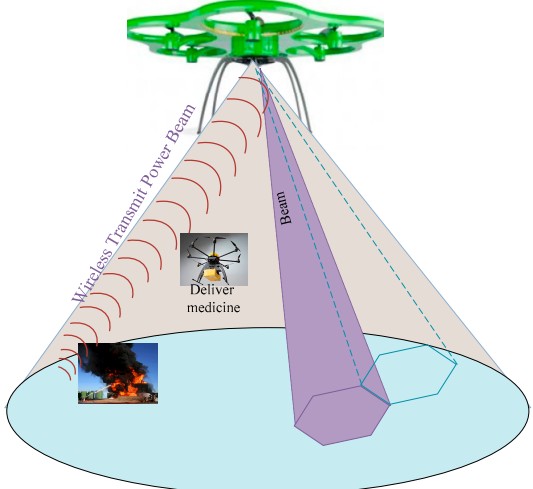

**Figure 5.** Drone for IoPST in emergency services and disaster recovery.

2.1.2. Probability of Line of Sight

The line of sight means the straight path from the signal source to a targeted location in three-dimensional space. Environmental impacts and elevation angle play a vital role in determining propagation features from a drone to IoT devices on the ground. For the design of a communication system, the transmitted signal should be received with sufficient signal strength without distortion. The probability distortion for LoS corresponding to each elevation angle in each propagation environment was calculated in [63]. The LoS probability from a smart drone to IoT devices on the ground will be used to estimate the coverage area and capacity, which allow IoT devices in smart environments to move around and gather data in the drone coverage area.

The primary application of the channel analyzer is the ability to optimize the performance of the smart drone rapidly. The altitude of a drone plays a vital role in determining the coverage area and received signal strength on the ground. Increasing the height of a drone will lead to enhancing the coverage area. Therefore, it will deliver services to a larger area and many users and devices will get services. Finding an optimized altitude is significant for maximizing the probability of coverage and small loss. If the height of the drone is h and the radius of the coverage area is R, then the distance between the drone and the ground receiver d is given in Equation (1):

$$d = \sqrt{R^2 + h^2}, \tag{1}$$

where the elevation angle $\theta$ of the drone concerning the user on the ground depends on the height of the drone and the coverage radius. This is given in Equation (2):

$$\theta = tan^{-1}(\frac{h}{R}).$$ (2)

The path loss depends on the carrier frequency $f_c$, the distance between drone and receiver on the ground d, and light speed c. The path loss is given by [64], as shown in Equation (3):

$$PL(dB) = \begin{cases} 20 \ \log(4\pi f_c d/c) + \varepsilon_{los}Los \\ 20 \ \log(4\pi f_c d/c) + \varepsilon_{nlos}NLoS \end{cases},$$ (3)

where $\varepsilon_{los}$ and $\varepsilon_{nlos}$ are the average additional losses to the free space loss. One of the most significant advantages of drone communication is LoS. Therefore, the LoS probability is crucial to correctly predicting signal attenuation among drones, IoT devices, robots or any objects on the ground. Prediction of accurate LoS probability determination helps to obtain more realistic path loss models. The elevation angle is the most critical parameter for predicting LoS probability. Plotting the probability of LoS depends on the parameter values of the environmental variables $\alpha, \beta, \gamma$, which are given different values in different areas based on [65]. The investigation of the probability of LoS condition based on [65] and the final formula of LoS probability is:

$$Plos = \prod_{n=0}^{m} [1 - e^{-\frac{[ht - \frac{(n+0.5)(ht-hr)}{m+1}]^2}{2\gamma^2}}]$$ (4)

$$m = r\sqrt{\alpha\beta} - 1$$ (5)

$$r = h \ tan(\theta),$$ (6)

where $ht$ and $hr$ represent the height of the receiver and transmitter, respectively; $r$ is the ground distance between the receiver and transmitter. However, plotting the resulting base on Equation (4) will smooth out because of the significant value of $ht$. Therefore, $Plos$ is considered to be calculated based on $\theta$ and environmental parameter impacts [63]. Holis et al. [63] directly suggested $Plos$ as:

$$Plos = C1 - (C1 - C2)/(1 + (\frac{\theta - C3}{C4})^{C5},$$ (7)

where $C1 \ldots \ C5$ are environments parameters are given in [63]. However, Hourani et al. [65,66] expressed the $Plos$ by sigmoid term concerning elevation angle $\theta$ as:

$$Plos = \frac{1}{1 + ae^{-b(\theta - a)}},$$ (8)

where a and b represent the S-curve parameters. Here, the Plos is easy to calculate and analytically flexible.

*2.2. IoT Devices for Public Safety*

IoT devices can play a vital role in enhancing the performance of a public safety network in terms of accuracy, efficiency, and predictability [45]. Furthermore, IoT devices have to analyze, aggregate information, and transmit without human intervention. This ensures the accuracy of received information and enhances the ability to anticipate crimes and other incidents. It is helpful to predict where crimes and natural disasters are more likely to occur. Figure 6 shows the classification of IoT for disaster management, which includes natural, manmade, service orientation, and post-disaster. Natural disasters include floods, earthquakes, landslides, etc.; however, the post-disaster focus is only

on saving victims in their locations. On the other hand, IoT for service-oriented disaster management has been discussed in [67–69].

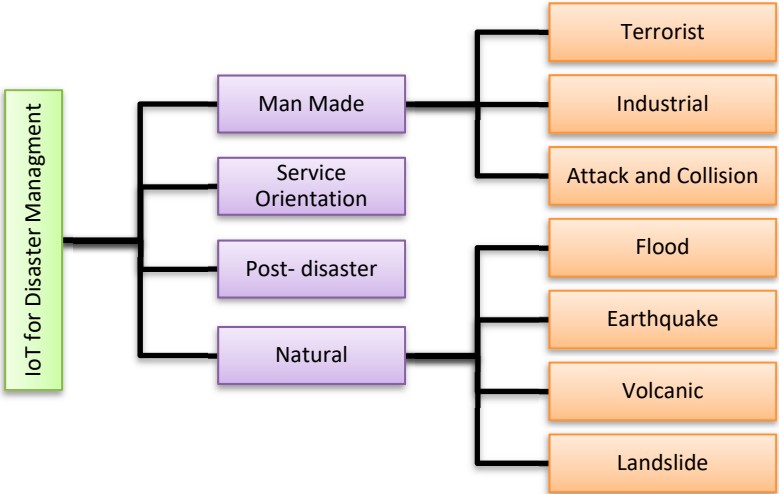

**Figure 6.** IoT-based disaster management classification.

## 2.3. Drones and IoT for Public Safety

Recently, drones have played a vital role in developing the next generation of wireless network and IoT. Therefore, they have attracted the attention of people who want to quickly develop drone applications that serve society, industry, government, and the environment. Furthermore, drones have a branch of sensors that can get real-time information everywhere they fly. Recently, a lot of connected sensors are being efficiently replaced by one drone that has the most essential features such as ease of deployment, flexible payload, programmability, and the ability to measure anything at any time anywhere. Drones can enable communication services while wireless communication networks are damaged during disaster [19]. They can also be used to gather information at a particular place and send the information to a control center in real time to aid with making the right decisions.

Drones offer several advantages over IoT stationary devices such as: they can be used for lengthy periods at any altitude, they have better resolution image quality, are low cost, have a rapid response, are capable of flying in any conditions, and are able to get closer to areas of interest. They represent a useful emerging technique for delivering wireless service to users or IoT devices on the ground, and have been investigated for their large coverage area and reduced power consumption [70].

## 3. Collaboration of Drones and Internet of Public Safety Things

IoPST wearable devices (i.e., sensors, cameras, other sensors) are ideal for first responders. The combination of a wireless sensor network and IoT is developed for a safety management system for tower crane groups [71]. Furthermore, wearable IoT devices present high potential for monitoring rescue and research teams during the disaster and communicating with the devices in the payload of the drone, as shown in Figure 2. The wearable IoPST devices connect over smart cities or a disaster area to allow first responders to share information with command posts and relevant parties, as shown in Figure 7. The drones play a vital role in collecting data from different devices during a disaster or scheduled event in smart cities. These connected devices help police and SAR teams identify when or where a crime or disaster happened. Therefore, police or SAR teams can make the right decisions and track down perpetrators. Here, drone technology is capable of carrying these devices and delivering services over a large coverage area, whereas connected devices are unable to send data a long distance due to their limited transmission power. The operational protocols of drones enable them to impact directly on the SAR team performance. SAR represents the most promising drone applications for

public safety. SAR processing can vary tremendously based on weather, environment, the experience of the searchers, and the location of the lost people.

Drones will send out the collected data in real time. Furthermore, they provide tactical assistance to increase public safety and officer safety in terms of situational awareness. Drones not only offer immediate care to those impacted by a crime/disaster but can also gather data that help cities rebuild after a disaster occurs. Furthermore, decision-making about public safety will be taken based on massive quantities of revel vent data collected by various devices connected to drones that are able to send data out in real time. Real-time data allow public safety authorities to quickly take action to keep people safe. The authorities will take real-time analytics based on real-time data received from a drone. Real-time analytics will help resolve public safety problems in time to help people before events/crimes occur. Therefore, the combination of drones and IoPST will contribute to many tasks and applications, as shown in Figure 8. The applications include police infrastructure, surveillance, equipment tracking, and emergency services.

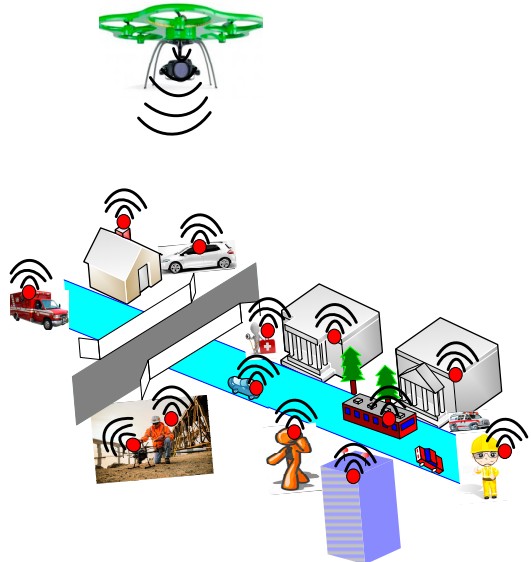

**Figure 7.** An illustration of drones and IoPST.

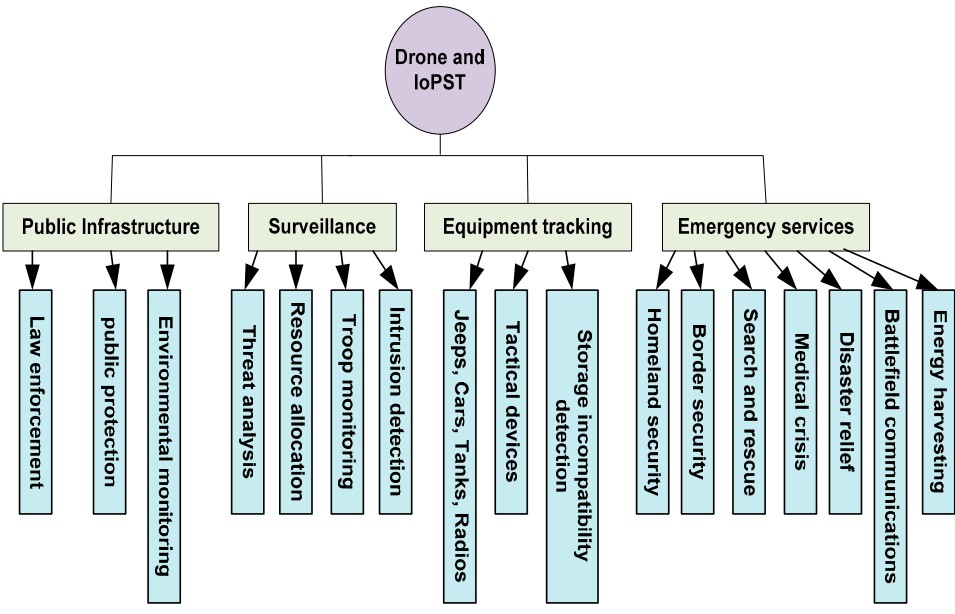

**Figure 8.** IoPST application and device proliferation.

## 4. Results and Discussion

The most important aspect of drones and the IoPST is the ability of these two technologies to form an effective and efficient communication network for public safety. Drones offer many benefits such as rapid deployment, LoS, reprogrammability, etc. Therefore, the network performance of the collaboration of drones and IoPST before, during, and after a disaster should improve greatly.

The simulation network is made using OPNET 14.5. The scenario consists of three drones and three SAR responders. A mobile ad hoc network (MANET) is proposed to connect the drones and responders to perform tasks efficiently. Also, the AODV routing protocol is considered for unicast and multicast routing services. Furthermore, the simulation time is 500 s. One drone is moving around closer to the events, as shown in Figure 2. This drone will capture an image of the event and share it with other drones and SAR on the ground for performing relevant actions according to the available data given by drone-2. Drone-1 and drone-3 are used to guide and deliver the supplies required by the SAR or people in a particular area. SAR-1, SAR-2, and SAR-3 are performing their duty by the guiding of drones and their wearable devices help drones to identify the location where they are needed. QoS parameters such as delay and throughput are considered for measuring the performance of the drones and SARs in particular events. The throughput and delay are measured for each device connected in a network, for testing joint performance during a disaster. Furthermore, the loss probabilities are calculated and simulated by a Matlab code.

The throughput increased with the increase of the packet size through all number of nodes. Thus, the difference between each number of nodes also increases during the increasing packet size in the different time, as shown in Figure 9. Throughput decreases when some wireless node increases. Throughput is the total data traffic in bits per second successfully received then forwarded to the drone. Each node has different throughput and Table 2 shows the minimum, maximum, average, and standard deviation (Std Dev). The maximum throughput is changed from 2814.4 bit/second in the case of drone-1 to 7908.8 bit/second in the case of SAR-1. Also, the average throughput is changed from a maximum of 4016.3 in the case of SAR-1 to 1827.2 in the case of drone-1.

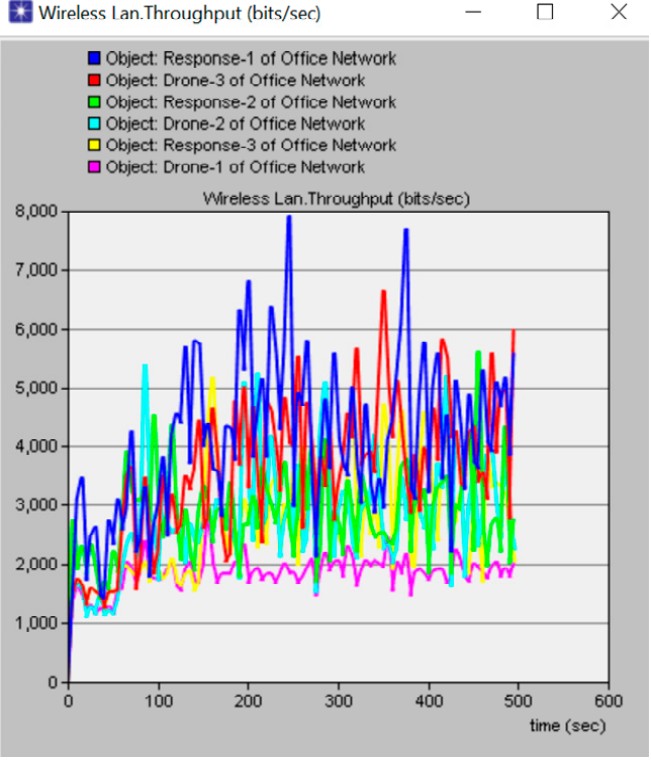

**Figure 9.** Throughput vs. time.

**Table 2.** Throughput for different nodes.

| Object | Minimum | Maximum | Average | Std Dev |
|--------|---------|---------|---------|---------|
| SAR-1 | 0 | 7908.8 | 4016.3 | 1368.3 |
| Drone-3 | 0 | 6648.0 | 3496.3 | 1243.9 |
| SAR-2 | 0 | 5624.0 | 2788.6 | 782.0 |
| Drone-2 | 0 | 5395.2 | 2722.4 | 963.7 |
| SAR-3 | 0 | 5180.8 | 2620.0 | 926.9 |
| Drone-1 | 0 | 2814.4 | 1827.2 | 330.3 |

The time delay will be affected by an increasing number of wireless nodes, and will decrease with an increase in packet size through all different number of nodes, as shown in Figure 10. In the case of minimum delay, SAR-2 is the lowest, but drone-2 has the highest minimum delay, which is 0.8 ms. However, drone-1 has no delay. Also, it is the same in the case of maximum and average delay for each node, as shown in Table 3.

**Table 3.** Delay for different node.

| Object | Minimum | Maximum | Average | Std Dev |
|--------|---------|---------|---------|---------|
| Drone-2 | 0.0008937 | 0.014345 | 0.0017257 | 0.0018513 |
| Drone-3 | 0.0010700 | 0.005018 | 0.0015526 | 0.0007544 |
| SAR-3 | 0.0010346 | 0.007813 | 0.0014888 | 0.0009098 |
| SAR-1 | 0.0010634 | 0.007173 | 0.0014134 | 0.0006074 |
| SAR-2 | 0.0002425 | 0.011812 | 0.0005631 | 0.0012831 |

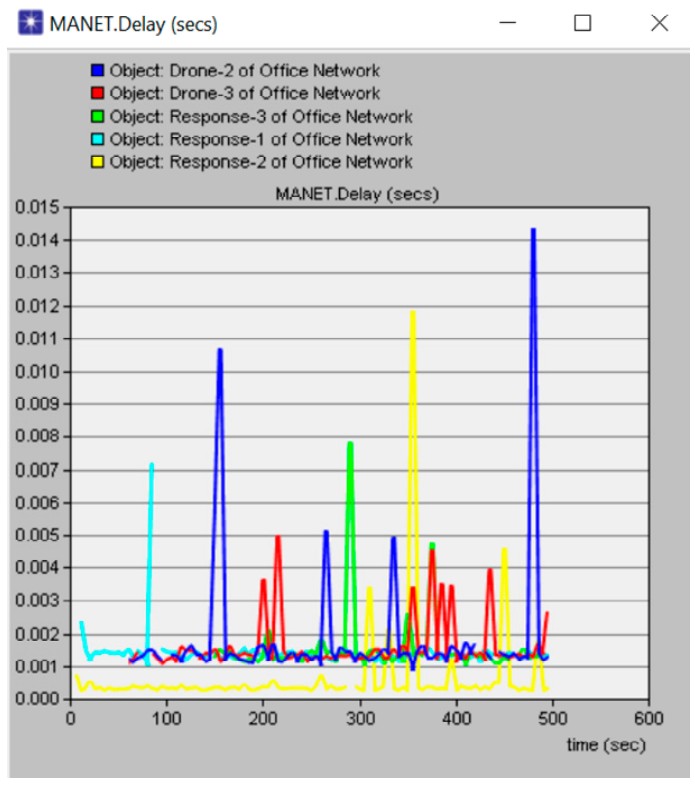

**Figure 10.** Delay for different nodes vs. time.

The LoS probability vs. elevation angle is depicted in Figure 11 for urban and suburban areas and dense urban area. Figure 11 shows the S curve, which is a direct approximation of ITU-R P.1410-2. However, the prediction is more accurate. Generally, the LoS probability increases when increasing the elevation angle. LoS probability reached a maximum at 30° elevations in the urban area. However, the LoS probability reaches a maximum at 60° and 70° in the case of suburban and dense urban areas, respectively.

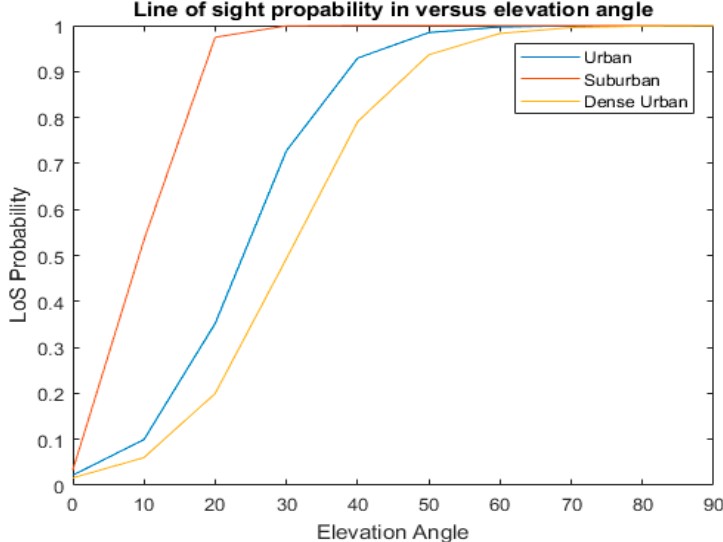

**Figure 11.** LoS probability of drone in different environments vs. elevation angle.

Figure 12 depicts the LoS probability based on a different elevation angle from the smart drone for different environments. The LoS probability changes according to the environment either an urban, suburban, or dense urban area. However, in urban areas the elevation angle should be small to keep the LoS probability small and the elevation angle should not be greater than 70°; otherwise, the connection will be lost. It is observed that the LoS probability shows similarities with the LoS presented in [16], but here the prediction was more accurate because of the estimation of the LoS probability at any location and elevation angle.

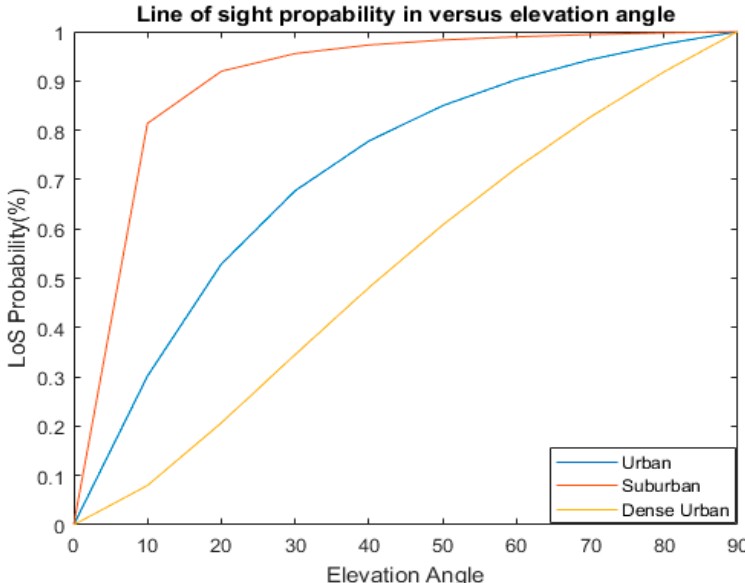

**Figure 12.** LoS probability with different elevation angle and environment vs. elevation angle.

## 5. Conclusions

Drone technology is increasingly being applied to improve first responses and public safety, due to the important features such as easy deployment, low cost, and ease of maintenance and operation. Drones also offer analytically sophisticated remote sensing solutions in SAR and medical supply delivery to remote locations, and so on. From this work, it is concluded that reliable public safety networks could be developed through the application of IoPST and drone technology. The drone-based IoPST would provide efficient connectivity services in the evacuation of affected people and can control crime much more conveniently with the interaction of relief, rescue, and police teams. Thus, the embedded system of IoPST and drones can be deployed during red alerts at unreachable sites for achieving data communication with a large coverage area. Hence, the collaboration of IoPST and drones would provide instant connectivity services with enhanced QoS parameters (i.e., delay and throughput) when there is a failure of ground-based wireless communication infrastructure.

**Author Contributions:** The presented work was carried out with the collaboration of all authors, who equally contributed.

**Funding:** This research was partially supported by Research Development Funds from Tsinghua University.

**Conflicts of Interest:** The authors declare no conflict of interest.

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
