# Peer review of "Collaboration of Drone and Internet of Public Safety Things in Smart Cities: An Overview of QoS and Network Performance Optimization"

_drones, doi:10.3390/drones3010013_

Round 1

Reviewer 1 Report

The authors present some results about the application of UAV-networks for public safety applications. The research topic is very interesting. However, the quality of the paper is in general too low.

1) The authors do not describe properly their approach. In fact, for the reader is very difficult to find the details of their approach. 

2) The simulations are not explained properly. Many details are missing about the simulations. 

The description of the scenarios.

Simulation time.

Number of UAVs.

Wireless Technology.

Etc. 

3) There important references about the topic that have not been included:

Evsen Yanmaz, Saeed Yahyanejad, Bernhard Rinner, Hermann Hellwagner, and Christian Bettstetter. Drone Networks: Communications, Coordination, and Sensing. Ad Hoc Networks, Special Issue on Advances in Wireless Communication and Networking for Cooperating Autonomous Systems, vol. 68, pp. 1–15, January 2018

https://www.sciencedirect.com/science/article/pii/S0140366416304315

https://www.mdpi.com/1424-8220/18/10/3571

https://www.sciencedirect.com/science/article/pii/S0167739X17325372

Author Response

Thank you very much for your invaluable comments. We have given our careful attention to the comments and have revised the manuscript accordingly.

Reviewer 2 Report

1. The manuscript is well written.
2. Introduction section is over explained and is quite unnecessary lengthy.
3. Author should include recent publications.
4. Simulation results are not clear. Aslo, simulation parameters must explained in detail.
5. Author need to analyse and explain in detail for figures from 8 to 11.
6. Also, author need to restructure the problem statement.
7. Conclusion part need to revise.

Author Response

Thank you very much for your invaluable comments. We have given our careful attention to your comments and have revised the manuscript accordingly to address your comments.

Reviewer 3 Report

In this paper, the authors study the network performance of the collaboration of internet of public 13 safety things and drone. The reviewer's comments are as follows:

1) The main novelty and contribution of the paper are not clear. In particular, which new aspects of drones in public safety applications are presented here?

2) There are many challenges and important issues related to drone communications which are not discussed in this paper. For example, 3D deployment, energy and flight time constraints, and interference management.

3) It is important to discuss appropriate types of drones for public safety applications.

4) The presentation of the paper should be significantly improved. In particular, the quality of figures and equations needs to be improved.

Author Response

(The authors gave the same response as above.)

Round 2

Reviewer 3 Report

The previous major comments have been clearly addressed. Nevertheless, the following minor comments need to be taken into account:

1) The literature review can be improved. In fact, there are some relevant studies on drone and Internet of Things that can be mentioned in the introduction. For instance, the following works can be considered:

[R1] “Unmanned aerial vehicle with underlaid device-to-device communications: Performance and tradeoffs,” IEEE Transactions on Wireless Communications, vol. 15, no. 6, pp. 3949–3963, June 2016. 

[R2] “Internet of Things for smart cities,” IEEE Internet of Things Journal, vol. 1, no. 1, pp. 22–32, Feb. 2014.

2) Please improve the quality of Figure 3b (drone), and proof read the paper to revise typos.

Author Response

(The authors gave the same response as above.)
